# Neighborhood-Level Influences and Adolescent Health Risk Behaviors in Rural and Urban Sub-Saharan Africa: A Systematic Review

**DOI:** 10.3390/ijerph18147637

**Published:** 2021-07-18

**Authors:** Stephanie Wiafe, Ariana Mihan, Colleen M. Davison

**Affiliations:** 1Department of Public Health Sciences, Queen’s University, Kingston, ON K7L 3N6, Canada; 12saw8@queensu.ca (S.W.); a.mihan@queensu.ca (A.M.); 2Department of Global Development Studies, Queen’s University, Kingston, ON K7L 3N9, Canada

**Keywords:** adolescent health, health risk behaviors, sub-Saharan Africa, neighborhood factors, health and place, urban health, rural health, health risk behaviours

## Abstract

The impact of neighborhoods on adolescent engagement in health-risk behaviors (HRBs), such as substance use and sexual activity, has been well documented in high-income countries; however, evidence from low and middle-income country settings is limited, particularly in sub-Saharan African (SSA) countries. Neighborhoods and communities in SSA continue to experience urbanization, epidemiologic transition, and the simultaneous presence of large populations living in rurality and urbanicity. This is a dynamic context for exploring adolescent health challenges. This review seeks to identify and summarize existing literature that investigates adolescent engagement in HRBs when compared across rural and urban neighborhoods across SSA. We performed searches using three electronic databases, targeted grey literature searches and scanned reference lists of included studies. Following dual-screening, our search yielded 23 relevant studies that met all inclusion criteria. These were categorized into six broad themes including studies on: (1) sexual risk taking, (2) injury-related, (3) violence, (4) eating and/or exercise-related, (5) substance use, and (6) personal hygiene. We found that neighborhood factors relating to accessibility and availability of health information and care impacted adolescent engagement in HRBs in rural and urban areas. Urbanization of areas of SSA plays a role in differences in engagement in HRBs between rural and urban dwelling adolescents.

## 1. Introduction

Adolescence is a critical period in the life cycle when rapid developmental changes take place, including physical growth, changes in societal roles and behaviors, and evolution in familial and interpersonal relationships [1,2]. During adolescence, individuals gain autonomy and there is a shift away from family-centered relationships to social relationships outside the home and family context [1,2]. Neighborhoods become a primary setting for social interaction, development and indeed for the determinants of health and illness of adolescents [3]. A growing body of evidence, particularly from high-income country (HIC) contexts, suggests that neighborhood factors are primary influences on adolescent health and well-being and that neighborhoods both directly and indirectly affect differential development trajectories during adolescence [3,4].

There are several reasons why neighborhoods are so influential for the health of adolescents. First, adolescents often spend significant amounts time during their day-to-day lives in the physical and social environments of their neighborhood [5]. Physical characteristics such as built structures, air and water quality, indoor and outdoor noise, greenspace, housing and waste removal systems directly affect health through the environmental exposures of neighborhood residents, including adolescents [6]. Additionally, a neighborhood’s social environment, with its unique social networks, existence or lack of support structures or levels of cohesion, for instance, have also been shown to greatly influence adolescent health. For example, social environments of neighborhoods affect adolescent self-identity, sense of safety, and esteem [6,7]. They also affect adolescent health behaviors, including those around substance use, sexual behaviors, risk taking and help seeking [8,9,10,11]. These impacts are significant because we know that adolescents who engage in health risk behaviors (HRBs) can experience lasting harmful effects, including the development of communicable as well as non-communicable diseases [12]. Health risk behaviors among young people are significant contributors to the global burden of disease [13]. In addition, while many of these neighborhood factors may affect adolescents’ immediate health and well-being, their influence can also be time-lagged, in that the influence of neighborhood factors may manifest later in life with effects continuing far into adulthood [1].

Sub-Saharan Africa (SSA) is recognized as one of the world’s fastest urbanizing regions in the world [14] creating the simultaneous presence of large metropolitan cities and rural villages [15]. In 2018, an estimated 472 million people were living in SSA, which is expected to double by the year 2042. Urbanization rates vary across the region, with southern countries in SSA having the fastest rate of urbanization and highest proportion of urban dwellers currently, followed by countries in western, central, and then eastern SSA [14,16]. Rapid development in SSA has also led to the rise of informal settlements in urban areas with more than half of the continent’s population living in these kinds of slums or shantytowns in extreme poverty [17]. While urbanization can promote economic growth and development, income inequality across socio-economic levels has been found to be greater in SSA’s urban cities, compared to its rural areas [17,18]. 

SSA is also a global region where adolescents aged 10–19 years make up a greater proportion of the population (23%) than the worldwide average (16%) [19]. Many countries in SSA have also undergone a population health epidemiological transition that is associated with changes in lifestyle factors such as access to diverse food sources, increases in tobacco use and changes in movement behaviors [20,21]. This has resulted in a dramatic balancing of disease burden between communicable, non-communicable, and injury causes [20,21]. The societal level changes associated with urbanization and changes in lifestyle factors have provided significant opportunities, but also health challenges for adolescents in SSA [22]. Neighborhoods and their effects on adolescent health and health behaviors have been documented and summarized in several HIC and in urban regions of low- and middle-income countries (LMICs) [3,23]. For instance, studies across the United States have found that adolescents dwelling in urban settings have demonstrated higher rates of illicit drug use [24,25,26,27]. In Scotland, adolescents living in urban areas were found to have poorer dietary habits compared to their rural-dwelling counterparts [28]. Evidence from Portugal found that urban-dwelling adolescents engage in more physical activity and less sedentary time compared to their rural counterparts. This was found to be due to neighborhood-level factors and their influences on active (versus sedentary) lifestyle practices, such as transport to and from school, namely active transport such as walking or bicycling versus sedentary transport such as taking the bus or driving [29]. Adolescents living in LMICs are increasingly engaging in HRBs that have been associated with social and physical environmental factors [30]. For example, a study assessing HRB among rural and urban-dwelling adolescents in Guatemala found urban adolescents engage in a number of HRBs, (including alcohol use and tobacco use) at higher rates compared to rural-dwelling adolescents. The authors posit that products of urbanization such as greater exposure to media and advertisements (for tobacco and alcohol) and easier access to substances in urban areas may explain these geographic differences [31]. Conversely a study investigating adolescent engagement in HRBs and urbanicity in Mexico found larger localities to be associated with a lower risk of tobacco and alcohol use, compared to rural localities. The authors here suggest that this difference may be associated with socially related neighborhood-level factors, such as attitudes and norms present in rural areas which normalize the use of alcohol and tobacco, in addition to a greater number of options for leisure activities available in urban areas, as compared to rural areas [32].

A summary of the differences in adolescent health risk behaviors between rural and urban neighborhood dwellers in sub-Saharan African has not yet been undertaken. Evidence about the broader context and determinants of adolescent health in the SSA region is limited, with much of the existing literature having a narrow focus on sexual and reproductive health issues affecting young people [33]. The documented effects of the significance of neighborhoods on adolescent engagement in HRBs in both LMICs and HICs is still in question. We were interested in investigating the effects of neighborhood-level influences on adolescents in SSA specifically, given the region’s significant population of adolescents and the current focus of health programming in the region on improving adolescent health [34]. We were interested in considering neighborhood effects within the changing contexts of urbanization and the epidemiological transition in SSA. Urban areas can be centers for advanced health care but can also create health problems [35]. Rural and non-rural youth can have vastly different health profiles and behaviors [36]. This study aimed to answer the following research question: in sub-Saharan Africa, what is the association between rural versus urban neighborhood characteristics and the health risk behaviors of adolescents? Objectives of this review included identifying neighborhood-level influences on adolescent health in rural and urban areas in addition to comparing adolescent engagement in HRBs in rural and urban areas in SSA. This review sought to identify and summarize existing literature that investigated adolescent engagement in HRBs in rural and urban regions of Sub-Saharan Africa in order to inform understandings of these contemporary, neighborhood-level health determinants for African adolescents and to ultimately inform health programming that targets improvements to adolescent health. 

## 2. Materials and Methods

This study is a systematic review that took place between July 2019 and March 2021 and was guided by the Preferred Reporting Items for Systematic reviews and Meta-Analyses (PRISMA) checklist (Appendix A) [37]. A protocol was developed and is unpublished.

### 2.1. Search Strategy

OVID Medline, Embase, and PsycInfo were searched using combinations of search terms and MeSH subject headings (Appendix B). The initial database searches were conducted on 6 August 2019. The search was re-run in March 2021 to capture any additional studies published between the initial search and March 2021. The database search strategy was designed in consultation with a health sciences librarian. Studies were exported from OVID into Covidence Systematic Review Software. Reference lists of included studies were scanned for additional studies A grey literature search was also conducted using Google and on websites relevant to our research question, including the World Health Organization, the United Nations, and the Africa Health Organization. 

### 2.2. Screening References

Studies were screened for eligibility in a two-stage process by two reviewers (S.W., A.M.), independently. First, titles and abstracts were screened and citations that did not meet established selection criteria were excluded. Next, both reviewers independently screened the full-texts of remaining studies for eligibility. Citations that were not eligible were removed. If reviewers did not agree, a third reviewer was consulted (C.M.D). 

### 2.3. Selection Criteria 

Studies were included in this review if they included adolescent participants (aged 10–19 based on the World Health Organization’s definition of adolescence) [38]; measured and/or described HRBs in the health risk behavior framework used by the Center for Disease Control and Prevention’s Youth Risk Behavior Surveillance System [39]. Studies had to include both rural and urban neighborhood settings. Reported results had to include data from at least one sub-Saharan African country. Observational quantitative, qualitative or intervention studies were included that were available in English full text. Citations were excluded if they were a review, an editorial, newspaper, or a form of popular media. Conference abstracts, commentaries and letters were also excluded due to a lack of detailed information. Studies were excluded if they did not include data for rural and urban neighborhoods for adolescents (10–19 years) in at least one SSA country.

### 2.4. Data Extraction and Analysis

Detailed information about each included study was recorded to identify and compare neighborhood-level influences on adolescent engagement in HRBs in rural and urban neighborhoods of sub-Saharan Africa. Neighborhood-level factors identified in each study were extracted and examined from study results and based on authors’ interpretation of their results. A data extraction form was used to collect the title, author(s), publication date, study setting, study type, sample size, participants age range, health risk behaviors investigated, and key findings including prevalence, odds ratios, relative risks (including *p*-values and/or confidence intervals) and qualitative data regarding adolescent HRBs in rural and urban neighborhoods of sub-Saharan Africa. Data extraction was performed by two authors (S.W., A.M). To calibrate and ensure consistency in extraction, both reviewers first independently extracted data from five studies and then compared extracted data and resolved discrepancies by joint review and consensus. In all cases, two authors assessed the articles and extracted the data. As heterogeneity exists in the type of studies, study settings, exposures and outcomes included in this review, a statistical meta-analysis was not appropriate. Evidence was summarized in table form and a narrative report of the findings was prepared. 

### 2.5. Quality Assessment

The Newcastle-Ottawa Quality Assessment Scale (adapted for cross-sectional studies) [40] was used as a tool for assessing the quality of the cross-sectional studies included in this review. The Newcastle-Ottawa Quality Assessment Scale evaluates studies based on three criteria: selection, comparability, and outcome on a scale of zero to ten, with zero being the lowest quality rating and ten being the highest. Cross-sectional studies were divided among two reviewers (S.W. and A.M) and were evaluated. The CASP Checklist for qualitative research studies, composed of three sections: validity, results and value, and encompassing ten checklist questions, was used to assess the quality of the qualitative study included in this review [41]. Quality assessments of each of the studies were validated by the other reviewer who established a scale of interpretation for the quality assessment rankings; a score of seven and above being methodologically strong, a score of five to seven indicating a moderate methodological strength and a score below five indicating a study with a high risk of bias. We did not exclude any studies based on quality.

## 3. Results

Our systematic search ultimately yielded 23 unique studies [42,43,44,45,46,47,48,49,50,51,52,53,54,55,56,57,58,59,60,61,62,63,64] which met all of our inclusion criteria (Figure 1). Studies excluded at the full-text stage were most frequently excluded on the basis of the following a priori exclusion criteria: not presenting rural–urban comparison of results specifically for adolescent-specific strata, for not comparing rural–urban populations at all, and/or for not focusing on any of the six HRBs that were deemed priorities in this review. Other reasons that studies were excluded not being an observational quantitative, qualitative or intervention study, not including adolescents or because the full-text article was unavailable.

### 3.1. Study Characteristics 

All studies took place in sub-Saharan Africa, with adolescent study participants. Of the 23 included studies, 21 (91%) were cross-sectional study designs [42,43,44,46,47,48,49,50,51,52,53,54,55,56,57,59,60,61,62,63,64] and two (9%) were qualitative studies [45,58]. Age ranges of study participants varied, with the youngest study participant being 10 years of age and the eldest being 19 years. Type of HRBs investigated varied among studies and six studies investigated more than one HRB (26%) [46,52,55,56,57,64]. Twelve studies investigated risky sexual HRBs (52%) [42,43,44,46,47,51,54,56,58,59,62,64], three studies investigated injury-related HRBs (13%) [53,55,61], one study investigated violence-related HRBs (4%) [55], two studies investigated HRBs related to personal hygiene (9%) [44,49], six studies investigated HRBs related to eating and/or exercise (26%) [45,49,50,52,57,60], and five studies investigated HRBs related to substance use (22%) [46,48,56,63,64]. Fifteen studies reported statistically significant results (*p* < 0.05) on urban and rural dwelling adolescent engagement in HRBs and/or neighborhood-level factors associated with adolescent engagement in HRBs (65%) [42,46,48,49,50,51,52,53,54,57,59,60,61,63,64]. Twenty-two of the 23 included studies (96%) relied on self-reported data with no additional form of verification. One study employed objective measures of adolescent engagement in HRBs in addition to self-reported data [49]. Twenty-one of 23 included studies (91%) investigated study populations comprised of both male and adolescent participants; two studies investigated a study population of adolescent females, exclusively [56,59]. The characteristics of each study are presented in Table 1.

### 3.2. Study Findings

A summary of the findings of each study are presented in Table 2. 

#### 3.2.1. Risky Sexual Behaviors

Of the 12 studies which investigated risky sexual behaviors among adolescents [42,43,44,46,47,51,54,56,58,59,62,64], seven (58%) reported a higher likelihood of rural dwelling adolescents engaging in risky sexual behaviors [42,44,51,54,58,59,62], while others reported a similar likelihood of engagement in a certain risky sexual behaviors when comparing rural and urban dwelling adolescents [44]. Five studies reported mixed results depending on the type of risky sexual behavior or the gender of participants [43,46,47,56,64]. Among these mixed results, common trends were seen for specific behaviors, such as condom use being less frequent among rural adolescents compared to urban adolescents [46,56,64]. Neighborhood-level influences associated with adolescent engagement in risky sexual behaviors included: sleeping in a different house from the household head, neighborhood-level and societal attitudes and norms surrounding adolescent sexual activity and AIDS prevention, structural barriers to accessing contraceptives in certain neighborhoods, feelings of stigma in the neighborhood, local perceptions surrounding contraceptives, differences in exposure to HIV education, socioeconomic status (SES), and religion.

#### 3.2.2. Injury Related Behaviors and Violent Behaviors

Among the three studies which investigated injury related HRBs [53,55,61], all reported higher or at least equal likelihoods of urban-dwelling adolescent engagement in injury related HRBs as compared to their rural counterparts. Neighborhood-level influences associated with adolescent engagement in injury related HRBs include a history of physical attacks and exposure to violence in the neighborhood, levels of socioeconomic deprivation (e.g., food insecurity common in the neighborhood), and presence of motor vehicles. One study which investigated violence related HRBs [55], reported mixed results depending on the type violence-related HRB.

#### 3.2.3. Risky Behavior Related to Personal Hygiene 

Of the two studies investigating personal hygiene related HRBs [52,57], both reported a higher proportion of rural adolescents engaging in health risk behaviors that might be negative for their physical hygiene [57]. Neighborhood-level influences included the accessibility and affordability of physical hygiene services (such as dental services) in particular neighborhoods.

#### 3.2.4. Risky Behaviors Related to Eating and/or Exercise 

Among the six studies that focused on HRBs related to eating and/or exercise [45,49,50,52,57,60] five reported an increased likelihood in urban-dwelling adolescents of engaging in HRBs related eating and/or exercise [45,49,50,52,57]. One study reported similar likelihoods between urban and rural adolescents [50] and an additional study reported mixed results depending on the type of HRB [60]. Neighborhood-level influences associated with adolescent engagement in HRBs related to eating/exercise included: level of access and availability of fast food vendors and tuck stops, modes of transportation utilized around neighborhoods (active transport versus sedentary transport), types of activities engaged in during leisure time (active versus sedentary), regional SES, and regional differences in nutrition habits between rural and urban neighborhoods.

#### 3.2.5. Substance Use 

Of the five, studies which investigated HRBs related to substance use [46,48,56,63,64], one study reported an increased likelihood of rural dwelling adolescents of engaging in HRBs related to substance use [48], two reported urban dwelling adolescents as engaging in substance use related HRBs at a higher proportion [56,64] and two reported mixed results depending on the certain HRB [46,63]. Neighborhood-level influences included: local cultural and social norms, attitudes surrounding HIV/AIDS prevention, regional SES, overall access to the internet and norms around watching televised football in particular neighborhoods. 

Overall, 12 studies (52%) were methodologically strong, with a quality assessment rating of 7 out of 10 or higher, indicating a low risk of bias. (Table 3) [42,43,45,49,52,53,58,59,60,61,62,63].

## 4. Discussion

This systematic review of neighborhood-level influences and adolescent engagement in HRBs in rural and urban regions of sub-Saharan Africa identified 23 relevant studies. These studies varied considerably in quality and types of HRBs investigated but were quite similar in study methodology and the number of HRB categories investigated. All studies took place in sub-Saharan Africa. In general, the methodological quality of included studies was moderate. Fifteen studies reported statistically significant results (*p* < 0.05) on urban and rural dwelling adolescent engagement in HRBs and/or neighborhood-level factors associated with adolescent engagement in HRBs.

We categorized studies by six types of HRBs adapted from the Center for Disease Control and Prevention’s Youth Risk Behavior Surveillance System [39]. We identified risk and levels of engagement in HRBs between rural and urban adolescent groups and we identified neighborhood-level influences (in both rural and urban settings) on adolescent engagement in all six categories of HRBs. 

Over half (52%) of the included studies in this review investigated adolescent engagement in risky sexual behaviors, over half of which (58%) reported that rural dwelling adolescents had a higher likelihood of engaging in risky sexual behaviors. We found that the neighborhood influences on adolescent engagement in risky sexual behaviors were largely related to local structural barriers to accessing knowledge and services related to sexual and reproductive health. Our findings are consistent with wider global literature that has identified that adolescents living in urban regions of SSA have lower rates of adolescent pregnancy and higher rates of contraceptive use compared to their rural counterparts [65]. Evidence on the rural–urban gap in sexual and reproductive health among adolescents in SSA shows that urban settings can offer better accessibility to sexual and reproductive health services such as access to contraceptives and family planning, compared to rural settings [65]. Adolescents living in urban areas of SSA can also have better access to sexual and reproductive health services as a result of different social norms concerning marriage and fertility [65]. In addition, wider literature aligns with our findings and indicates major and persistent inequalities in sexual and reproductive health between rural dwelling and urban dwelling adolescents in SSA due, in part, to structural barriers to accessing sexual and reproductive health services [65]. In addition to access, wider literature also suggests that public health facilities in rural areas of SSA, which often provide sexual and reproductive services, such as contraceptives and sexual and reproductive health education, face barriers to providing care due to factors associated with their rurality and economic capacity [66]. There is a need to prioritize and address the disparities in sexual and reproductive health service delivery in rural and urban areas in SSA through community-level approaches that increase the availability and accessibility of sexual and reproductive health services for adolescents that consider local contexts as well as social and structural barriers. In a similar way, findings related to neighborhood-level influences on physical hygiene HRBs also indicated structural and economic barriers in rural areas, such as a lack of accessibility and affordability of dental health services [52]. Recent evidence on removing barriers to practicing hygiene in Southern Africa in the wake of the COVID-19 pandemic, is in alignment with our findings, identifying structural and economic barriers, such as a lack of a physical presence of hygiene facilities in rural communities and a limited infrastructure to support physical hygiene facilities and practices [67]. 

Exposure to violence was found to be a common neighborhood-level influence on adolescent engagement in health risk behaviors that were linked to injuries and violent behavior. This is consistent with the broader evidence base, which identifies exposure to violence as a risk factor for youth perpetration of violence [68]. Other neighborhood influences on engagement in injury related HRBs included levels of neighborhood socioeconomic deprivation, lack of social cohesion, and the presence and access to motor vehicles. All three studies in our review that focused on health risk behaviors and injury reported higher likelihoods of urban adolescent engagement in these behaviors. These findings are in alignment with evidence regarding the association between the effects of urbanization and urban areas in SSA and an increase in experiences of crime and violence [17]. A review published by the United Nations Development Programme on armed violence and urbanization in Africa indicates that rapid urban development in Africa has led to informal settlements such as slums and shantytowns in urban areas, which experience heightened exposures to and risk of violence and risk factors for violence, including factors identified in our study such as poverty, socioeconomic deprivation, and decreased social cohesion [17]. Our findings and the existing evidence base may suggest that sustainable urban planning and public safety programming and policies, which address the root causes of urban and urbanization-related violence such as socioeconomic deprivation, may also address urban dwelling adolescent engagement in injury and violence related HRBs. 

Five of the six of the studies included in this review which focused on adolescent engagement in HRBs related to eating and/or exercise reported an increased likelihood of urban adolescent engagement as compared to rural engagement. The other study reported mixed results depending on the specific risk behavior. Neighborhood-level factors on engagement in this category of HRBs were largely related to factors directly related to urbanization, such as usage of sedentary transport and access and availability of fast foods. This again highlights the effects of urbanization and urban neighborhoods on adolescent HRBs. It is widely agreed that urbanization of cities across SSA have come with increased consumption of fast foods and passive (as opposed to active) leisure time leading to health threats of obesity and other NCDs [69].

Studies included in this review also reported that differences in cultural and social norms between rural and urban neighborhoods influence adolescent engagement in HRBs, specifically those related to substance use. Traditions around these behaviors, such as the patterns and acceptability of smokeless tobacco use, may differ in rural and urban areas [26,27]. This coupled with differences in access to and availability of substances in different areas may explain the mixed results. 

The identified studies in this review vary in quality; although most were found to be adequate in quality, it is important to acknowledge that nine studies were of moderate quality and two were of low quality, and thus suffered limitations relating to their validity and reliability, including the use of measurement tools that have not been validated. It is important to consider these limitations when interpreting the results. The quality assessment took into account the exposure measurement and outcome assessment, including whether measurement tools were validated and if the outcome assessment was based off-self-reported data. The vast majority of the identified studies relied on self-reported data to measure adolescent HRBs, thus influencing the quality score of these studies. Self-reporting is conventional for studies examining HRBs, and self-report data are vulnerable to response and social desirability biases and potential misclassification, which can threaten the validity of the findings. However, previous studies investigating the comparison between objective measures and self-reported data of risk behaviors among adolescents indicate that self-reported data can also be valid [70,71]. A study assessing the reliability of the bi-annual United States Centre for Disease Control Youth Risk Behavior Survey, which collects national self-report data from American youth, found that though the survey collects self-reported data, the results indicate that youth reported on their health risk behaviors reliably over time [72]. There is a possibility that our review missed some relevant studies given the strict and detailed study selection criteria. For example, studies may have been excluded from the review if they did not stratify data by age and/or by urban and rural dwelling or if they were written in a language other than English. This review also did not investigate peri-urban settings in SSA. 

The results of our review are consistent with the evidence base on adolescent HRBs and neighborhood-level influences in sub-Saharan Africa and globally. Results of our review can be used to inform public health programming and policy to address the neighborhood-level barriers and facilitators to adolescent health and the risk factors for adolescent engagement in HRBs. Community-level approaches to improving access to health services for adolescents in rural communities and public health informed urban planning to reduce the negative effects or urbanization on adolescents in urban communities are warranted. Our review has identified specific neighborhood-level factors and how these are associated with adolescent engagement in HRBs. These details can help inform healthcare decision-making and support evidence-based approaches to improving the health and health behaviors of adolescents in both rural and urban neighborhoods. Our review has also identified areas for future research. First, given the effects of urbanization on adolescent engagement in HRBs in urban and rural settings, further research is needed to assess the relationship between aspects of peri-urban settings and adolescent health behaviors. Second, to inform evidence-based public health measures to address adolescent engagement in HRBs in the region, future research should investigate the effectiveness and applications of HRB public health interventions, such as targeted approaches to primary, secondary, and tertiary prevention of HRBs among adolescents which consider urbanicity and rurality. Lastly, studies investigating adolescent HRBs with self-reported data could employ biological verification and/or record linkage or longitudinal designs where appropriate and possible, to decrease the risk of bias.

## 5. Conclusions

Adolescence is a critical period during the life cycle, where engagement in HRBs can have immediate as well as lasting harmful effects. In sub-Saharan African countries, neighborhood contextual factors, including structural factors relating to accessibility to and availability of health information and medical care services, were found to be associated with adolescent engagement in HRBs in both rural and urban areas. Studies also indicate that urbanization in SSA may also be playing a role in creating differences in engagement in HRBs between rural and urban dwelling adolescents. Overall, the existing literature indicates that urban-rural status and neighborhood-level factors should continue to be considered when studying the determinants of adolescent health risk behaviors. Place-based interventions exist and should be further explored to support more positive behaviors among youth in sub-Saharan Africa.

## Figures and Tables

**Figure 1 ijerph-18-07637-f001:**
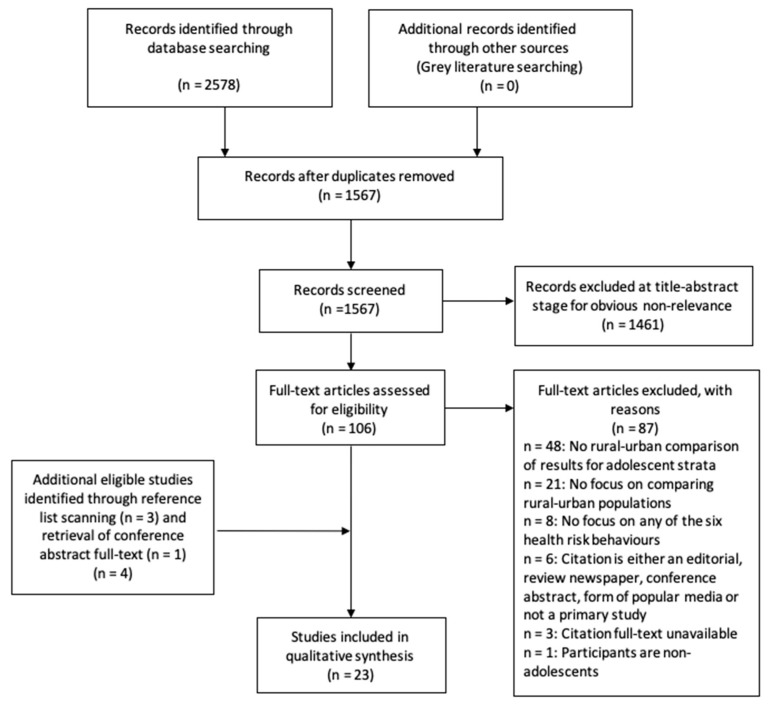
PRISMA diagram.

**Table 1 ijerph-18-07637-t001:** Included study characteristics and key findings.

No.	Title	Year	First Author	Adolescent Participants Age-Range (Years)	Study Design	Adolescent Sample Size (n)	Health Risk Behavior Topic Area(s)
[42]	Risky sexual behavior among orphan and non-orphan adolescents in Nyanza Province, Western Kenya	2013	Juma, M.	14–17	Cross-sectional	546	Risky Sexual Behaviors
[43]	Predictors of intention to be sexually active among Tanzanian school children	1996	Klepp, K.I.	10–17	Cross-sectional	2026	Risky Sexual Behaviors
[44]	Sexual behavior, reproductive health and contraceptive use among adolescents and young adults in Mbale District, Uganda	1994	Agyei, W.K.	15–19	Cross-sectional	1357 (aged 15–24)	Risky Sexual Behavior
[45]	“I eat to be happy, to be strong, and to live.” Perceptions of rural and urban adolescents in Cameroon, Africa	2007	Dapi, L.N.	12–15	Qualitative	15	Risky Behavior related to Exercise/Eating
[46]	AIDS-related knowledge, attitudes and behavior among adolescents in Zambia.	2005	Slonim-Nevo, V.	10–19	Cross-sectional	3360	Risky Sexual Behavior and Substance Use
[47]	Sexual behavior is riskier in rural than in urban areas among young women in Nyanza province, Kenya	2004	Voeten, H.A.	15–19	Cross-sectional	213	Risky Sexual Behaviors
[48]	Prevalence and correlates of smokeless tobacco use among grade 8–11 school students in South Africa: a nationwide study	2014	Reddy, P.S.	10–17	Cross-sectional	10,270	Substance Use
[49]	Effect of urbanization on objectively measured physical activity levels, sedentary time, and indices of adiposity in Kenyan adolescents	2012	Ojiambo, R.M.	12–16	Cross-sectional	200	Risky Behavior related to Eating/Exercise
[50]	Dietary habits and eating practices and their association with overweight and obesity in rural and urban black South African adolescents	2018	Sedibe, M.	10–15	Cross-sectional	3490	Risky behavior related to Eating/Exercise
[51]	Sexual behavior of in-school adolescents in Osun State, Southwest Nigeria: a comparative study	2014	Sabageh, A.O.	10–19	Comparative cross-sectional	760	Risky Sexual Behavior
[52]	Socio-demographic disparity in oral health among the poor: a cross sectional study of early adolescents in Kilwa district, Tanzania.	2010	Mashoto, K.O.	10–19	Cross-sectional	8609	Risky Behavior related to Eating/Exercise & Risky behavior relating to Personal Hygiene
[53]	Prevalence and correlates of suicidal behavior among adolescents in southwest Nigeria.	2008	Omigbodun, O.	10–17	Cross-sectional	1429	Injury-related Behaviors
[54]	Contraceptive use among in and out-of-school adolescents in rural southwest Uganda.	2006	Batwala, V.K.	15–19	Cross-sectional	720	Risky Sexual Behaviors
[55]	Injury-related behavior among South African high-school students at six sites.	2006	Flisher, A.J.	13–16	Cross-sectional	4563	Injury-related Behaviors and Violent Behaviors
[56]	STI-prevalence and differences in social background and sexual behavior among urban and rural young women in Uganda	2010	Darj, E.	12–19	Cross-sectional prospective	592	Risky Sexual Behaviors, Substance use
[57]	Oral hygiene and sugar consumption among urban and rural adolescents in Ghana	2000	Blay, D.	14–18	Cross-sectional	504	Risky Behavior related to Eating/Exercise & Risky behavior related to Personal Hygiene
[58]	Risky sexual behaviors among adolescent undergraduate students in Nigeria: does social context of early adolescence matter?	2020	Odii, A.	16–19	Qualitative	24	Risky Sexual Behaviors
[59]	Why too soon? Early initiation of sexual intercourse among adolescent females in Ethiopia: Evidence from 2016 Ethiopian demographic and health survey	2020	Turi, E.	15–19	Cross-sectional	3881	Risky Sexual Behaviors
[61]	Factors associated with depressive symptoms and suicidal ideation and behaviors amongst sub-Saharan African adolescents aged 10–19 years: cross-sectional study	2020	Nyundo, A.	10–19	Cross-sectional	7662	Injury-related Behaviors
[60]	Adolescents’ food habits and nutritional status in urban and rural areas in Cameroon, Africa	2005	Nzefa Dapi, L.	12–15	Cross-sectional	46	Risky behavior related to Eating/Exercise
[64]	Sexuality among adolescents in rural and urban South Africa	2006	Peltzer, K.	16–17	Cross-sectional	800	Risky Sexual Behaviors and Substance Use
[62]	Incidence and Predictors of Adolescent’s Early Sexual Debut after Three Decades of HIV Interventions in Tanzania: A Time to Debut Analysis	2012	Mmbaga, E.J.	16–19	Cross-sectional	316	Risky Sexual Behaviors
[63]	Prevalence and risk factors for initiating tobacco and alcohol consumption in adolescents living in urban and rural Ethiopia	2019	Getachew, S.	13–19	Cross-sectional	3967	Substance Use

**Table 2 ijerph-18-07637-t002:** Included study key findings.

No.	Main Findings (Neighborhood Effects on the Health Risk Behaviors)
[42]	Adolescents living in urban areas were less likely to have had sex compared to rural adolescents ^1^. Urban adolescents were less likely to engage in transactional sex and more likely to have used a condom at last sex, compared to their rural counterparts (not statistically significant). The authors report sleeping in a different house from the household head and living in a rural neighborhood associated with an increased likelihood of adolescent sexual encounters.
[43]	In the Kilimanjaro Region, the prevalence of adolescent sexual behavior was higher among urban males compared to rural male adolescent behavior. In the Arusha Region, adolescent sexual behavior was higher among rural males compared to urban males. Female adolescent sexual behavior was higher among rural participants in both the Kilimanjaro Region and in the Arusha Region. Neighborhood factors: within both regions, both males and females in rural areas held attitudes and norms more accepting sexual activity compared to urban adolescents.
[44]	Rural adolescents had slightly higher proportions of ever having sexual intercourse, compared to urban adolescents. Both rural and urban adolescents had the same median age at first intercourse. Urban adolescents reported a higher proportion of currently and ever using condoms. Neighborhood factors related to use of contraceptives include a higher proportion of rural females reporting not using contraceptives due to not being able to get them, compared to their urban counterparts; however, mixed findings were reported for males.
[45]	Rural adolescents ate less fast food than their urban counterparts due to low or no availability and fewer vendors. Urban adolescents reported having better food availability (though not always affordable), compared to rural adolescents. The authors reported that urban adolescents were more attracted to fast food due to accessibility through vendors on school yards, which are less prevalent among rural adolescents.
[46]	A higher proportion of rural adolescents traded sex for food or money ^2^, had sex while drunk or on drugs without a condom ^1^, had vaginal or anal sex without a condom, and injected needles ^2^ in the past 2 months compared to urban adolescents. Urban adolescents reported slightly higher proportions of having sex while drunk with a condom. Rural adolescents reported slightly higher proportions of sex while on drugs with a condom. Rural adolescents reported higher proportion of vaginal sex both with ^2^ and without a condom ^2^ and higher proportions anal sex both with ^2^ and without a condom ^2^. Adolescents living in rural areas engaged in more high-risk behaviors. Neighborhood-related factors included: urban adolescents had more positive attitudes towards prevention of AIDS. It was also reported that adolescents who came from lower SES engaged in higher risk behaviors.
[47]	The median lifetime number of partners for urban and rural adolescent men were the same; however, the inter-quartile range for rural men was higher compared to urban men. The median lifetime number of partners for urban adolescent women 15–19 was higher for rural adolescent women compared to urban adolescent women and the inter-quartile range for rural women was higher compared to urban women. The authors report that differences in sexual behavior between urban and rural adolescent women could not be explained by sociodemographic (neighborhood) differences.
[48]	Rural students had significantly higher odds of smokeless tobacco use compared to urban students ^3^. Students in upper SES schools had significantly lower odds than those from lower SES schools of using smokeless tobacco ^3^.
[49]	Adolescent rural males and females had less daily sedentary time compared to their urban counterparts. This result was statistically significant between rural and urban males ^1^. The authors reported differences in the leisure time activities between urban and rural adolescents, where urban adolescents engaged in more sedentary activities, while rural adolescents were involved in activities such as household chores that required them to be physically active. All rural adolescents used active transport (no motor transport) to get to get to school, in comparison to their urban counterparts where half used motorized (sedentary) transport to get to school.
[50]	The frequency of fast food consumption ^3^, mean number of fast food items consumed ^3^, mean number of snacks consumed whilst watching television ^3^, frequency of tuck shop purchases and mean number of items purchased at tuck stops ^3^ in the previous week, were higher among urban adolescents compared to rural adolescents aged 13 and 15. Frequency of snacks consumed whilst watching television was higher among urban adolescents aged 13 year old compared to their rural counterparts ^3^; however, was similar between urban and rural adolescents aged 15 years old. The authors report a likelihood of greater availability of snacks within urban adolescents’ homes due to SES factors.
[51]	A higher prevalence of rural adolescent men and women had already had sex at the time of the study compared to urban adolescent men and women ^1^. Half of the urban adolescents who were sexually active reported using a condom during their first intercourse compared with their rural counterparts, of whom only a quarter used a condom ^2^. Urban adolescents had a higher prevalence of ever using a condom compared to rural adolescents (finding not statistically significant). A higher proportion of urban adolescents reported using a condom the last time they engaged in sexual intercourse, compared to rural adolescents (not statistically significant). The authors reported that these findings in this study contradict what is expected due to the lifestyle of urban dwellers.
[52]	Urban adolescents had a higher sugar intake compared to their rural counterparts ^3^. In comparison to urban adolescents, a higher proportion of rural adolescents never/seldom engaged in tooth brushing ^3^ and did not use fluoridated toothpaste ^3^. The authors reported accessibility, affordability and structural barriers to dental services as reasons for rural adolescents’ lack of utilization of dental services
[53]	Suicidal ideation was similar among rural and urban adolescents. Suicidal attempts were higher among urban adolescents compared to their rural counterparts. ^1^ Being urban dwelling compared to rural dwelling was a significant predictor of suicide attempt ^1^. Neighborhood-level factors significantly associated with suicidal ideation include socioeconomic deprivation including food insecurity. Other factors that were significant predictors of suicidal ideation and attempts included sexual abuse, physical attack and fight involvement.
[54]	Rural in-school adolescents were less likely to use contraceptives compared to their urban in-school counterparts. For in school adolescents, place of residence was not a statistically significant predictor of contraceptive use. For out-of-school adolescents, place of residence was a significant predictor of contraceptive use; adolescents living in urban areas were more often contraceptive users compared to who resided in rural areas ^1^. Lack of access/availability of commonly used contraceptives and stigma were reported as barriers to out-of-school users ^1^
[55]	The prevalence of being a front passenger with no seatbelt, being a passenger in a car with an intoxicated driver, or carrying a knife to school was higher among urban adolescents compared to the rural adolescents. The prevalence of bullying others was higher in the rural region among adolescent men compared to the urban regions; however, results were mixed among adolescent women. The authors reported that neighborhood factors influencing the tendency for motor-vehicle related risk behaviors to be lower in rural areas is likely related to access.
[56]	A higher proportion of urban women reported smoking and drinking alcohol compared to their rural counterparts. Age at sexual debut was similar between urban and rural women. Condom use for first and most recent intercourse as well as constant condom use was more common among urban women compared to rural women. Rural women were using less-safe methods of contraception in comparison to urban women. A greater proportion of urban women reported having a new sexual partner within the last 3 months and reported having more than one sexual partner within the last six months more often than rural women. A higher proportion of rural, compared to urban women reported having sex in exchange for money or gifts. The authors cite a study which noted that in rural southwest Uganda, the exchange of sex for money and gifts is considered common within sexual partnerships. The authors report that the low rate of condom use among the rural women may be due to their higher rate of marriage.
[57]	Consumption of sugary snacks and drinks was significantly higher among urban adolescents compared to rural adolescents ^1^. Prevalence of tooth brushing was similar among rural and urban adolescents. The proportion of urban adolescents’ daily toothpicks use was higher compared to rural adolescents.^1^ The authors cite a study reporting differences in consumption of sugary drinks and snacks is likely due to due to region differences in nutrition.
[58]	Adolescents who were raised in rural environments in their early adolescents were described to be involved in risky sexual behaviors in university, such as being less likely to use condoms, due to having less exposure to sex education and more influence by misinformation and superstitions, compared to those raised in urban environments. Adolescents raised in religious environments were less likely to engage in risky sexual behavior.
[59]	In comparison to urban adolescents, rural adolescents had greater odds of early sexual initiation ^1^; however, this association did not remain significant when potential confounding factors were adjusted for. Neighborhood factors influencing the likelihood of engaging in early sexual intercourse included poor wealth status ^1^
[61]	Adolescents in Ibadan (urban setting) had the highest prevalence of suicidal behavior, suicide attempts, suicidal ideation, suicide plans, and suicide attempts. Harar (urban setting) had the second highest prevalence rates of suicidal behavior. Neighborhood factors positively associated with suicidal behavior included poor access to health care (Harar, Kersa, Dar es Salaam, Ningo Prampram, and Iganga/Mayuge) and food insecurity (Harar ^3^ and Ibadan ^2,^ Kersa, Ningo Prampram, and Iganga/Mayuge), and exposure to violence (Ibadan ^3^ Harar, and Ningo Prampram).
[60]	Urban adolescents high a significantly higher frequency of junk food consumption compared to rural adolescents ^2^. The authors reported that there is a greater availability and accessibility junk food in the urban area, such as through a multitude of vendors selling cheap junk food on school yards. However, urban adolescents also had a higher frequency of vegetable intake compared to rural adolescents.^3^
[64]	Urban adolescents had significantly more often engaged in sexual experiences such as oral sex and anal intercourse compared to rural adolescents ^3^. Urban adolescents had significantly greater number of sexual partners with whom compared to rural adolescents.^3^ Urban adolescents were significantly more likely to report being under the influence of drugs/alcohol as a reason for first sexual intercourse. ^3^ Urban adolescents reported more consistent condom use than rural adolescents ^1^. Urban adolescents were significantly more likely to report smoking/snuffing, drinking alcohol, and regularly using cannabis. ^3^ There was less prohibition/prevention through familial or societal controls of the sexuality of urban adolescents. The authors report that this influenced the higher rate of consistent condom use among urban adolescents. The study reported indications of egalitarian standards among urban youth (i.e., sexual initiative of women was more accepted among urban vs rural youth and urban female adolescents more frequently initiated first sex, compared to female rural adolescents.)
[62]	Growing up in an urban area was associated with a lower risk of early sexual debut. The authors reported that adolescents from urban regions may have a greater likelihood of exposure to HIV educational initiatives in comparison to their rural counterparts.
[63]	Residing in an urban region was associated with ever trying smoking ^2^. A slightly greater proportion of urban adolescents reported current smoking; however, the results were not significant. A slightly greater proportion of urban adolescents, compared to rural, reported ever drinking alcohol; however, the results were not significant. In contrast, a greater proportion of rural adolescents reported current regular alcohol use ^3^. Access to internet was significantly ^3^ associated with cigarette smoking and with alcohol use. Alcohol use was positively related to how much students watched televised football.

^1^*p* < 0.05, ^2^
*p* < 0.01, ^3^
*p* < 0.001.

**Table 3 ijerph-18-07637-t003:** Quality assessment of studies.

No.	First Author	Quality Assessment Rating	Rating Breakdown
[42]	Juma, M.	7/10	Selection: 4/5 Comparability: 1/2 Outcome: 2/3
[43]	Klepp, K.I.	7/10	Selection: 4/5 Comparability: 1/2 Outcome: 2/3
[44]	Agyei, W.K.	4/10	Selection: 2/5 Comparability: 1/2 Outcome: 1/3
[45]	Dapi, L.N.	9/10	Validity: 5/6 Results: 3/3 Value: 1/1
[46]	Slonim-Nevo, V.	5/10	Selection: 2/5 Comparability: 1/2 Outcome: 2/3
[47]	Voeten, H.A.	5/10	Selection: 2/5 Comparability: 1/2 Outcome: 2/3
[48]	Reddy, P.S.	6/10	Selection: 3/5 Comparability: 1/2 Outcome: 2/3
[49]	Ojiambo, R.M.	7/10	Selection: 3/5 Comparability: 1/2 Outcome: 3/3
[50]	Sedibe, M.	5/10	Selection: 2/3 Comparability: 1/2 Outcome: 2/3
[51]	Sabageh, A.O.	6/10	Selection: 3/5 Comparability: 1/2 Outcome: 2/3
[52]	Mashoto, K.O.	8/10	Selection: 5/5 Comparability: 1/2 Outcome: 2/3
[53]	Omigbodun, O.	7/10	Selection: 4/5 Comparability: 1/2 Outcome: 2/3
[54]	Batwala, V.K.	5/10	Selection: 2/5 Comparability: 1/2 Outcome: 2/3
[55]	Flisher, A.J.	5/10	Selection: 2/5 Comparability: 1/2 Outcome: 2/3
[56]	Darj, E.	4/10	Selection: 2/5 Comparability: 1/2 Outcome: 1/3
[57]	Blay, D.	6/10	Selection: 3/5 Comparability: 1/2 Outcome: 2/3
[58]	Odii, A.	9/10	Validity: 5/6 Results: 3/3 Value: 1/1
[59]	Turi, E.	7/10	Selection: 3/5 Comparability: 2/2 Outcome: 2/3
[61]	Nyundo, A.	8/10	Selection: 4/5 Comparability: 2/2 Outcome: 2/3
[60]	Nzefa Dapi, L.	8/10	Selection: 5/5 Comparability: 0/2 Outcome: 3/3
[64]	Peltzer, K.	5/10	Selection: 3/5 Comparability: 0/2 Outcome: 2/3
[62]	Mmbaga, E.J.	7/10	Selection: 3/5 Comparability: 2/2 Outcome: 2/3
[63]	Getachew, S.	8/10	Selection: 4/5 Comparability: 2/2 Outcome: 2/3

## Data Availability

Not applicable.

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
