# Peer review of "Neighborhood-Level Influences and Adolescent Health Risk Behaviors in Rural and Urban Sub-Saharan Africa: A Systematic Review"

_ijerph, 2021, doi:10.3390/ijerph18147637_

Round 1
Reviewer 1 Report
This is an interesting, well-written, and useful manuscript. I have only one comment for the authors to consider.
In table 2, most studies show selection, comparability, outcome. But a couple show validity, results, value. Why is this? Should it be corrected?
Author Response
Reviewer 1: In table 2, most studies show selection, comparability, outcome. But a couple show validity, results, value. Why is this? Should it be corrected?
Response: We used two tools for assessing the quality of the included studies, one appropriate for cross-sectional studies and the other for qualitative studies. The two studies that show validity, results and value are qualitative studies and were assessed with the qualitative tool. The remaining studies are cross-sectional and were assessed with the cross-sectional evaluation tool. We have added to the methods section a line clarifying that the tool used to assess qualitative studies encompassed three sections: validity, results and value.
Reviewer 2 Report
International journal of Environmental Research of Public Health
Neighbourhood-level influences and adolescent health risk behaviors in rural and urban sub-Saharan Africa: A systematic review
The aim of the paper was to identify and summarize existing literature, investigating adolescent health risk behaviors between rural and urban neighbourhood dwellers in sub-Saharan African. The author used systematic review according The PRISMA statement (Moher et al 2009). Twenty-three articles were included in the quantitative synthesis.
Suggestion: Major Revision
- PRISMA-checklist should be included as an additional file.
- Figure 1. Revise figure 1 according to the PRISMA statement. What about the grey literature?
- Table 1. Huge table. Structure the table. For example, the country name is mentioned twice in the table. The Main-finding should be narrow down to only regarding HRB. Perhaps divide table 1 to two tables to clarify the main findings.
- Appendix A: Search strategy for each database is needed, please also make the appendix clearer, it’s not readable.
- Rationale & Objectives: Why is it important to compare HRB urban rural in sub-Sahara? The aim needs to be clearer with one or two research questions. Is it similarities or/and differences?
- Methods: Clarify the score of the quality assessment, what does the score 7 means? Risk of bias?
- Please describe more about the differences between included studies and excluded studies.
Line 177-1978, “fifteen studies reported statistically significant…” of what?
To clarify the study characteristics – a table or diagram would improve the results section.
Structure the results section to make it more readable- the synthesis of results.
- Discussion
Summary of results: Line 254-257, this is not seen in the results section or is a reference missing.
Limitations. Self-reported-data and HRB and adolescents needs to be discussed more and self-reported data in the included study.
Risk of bias. Why did you choice not to exclude articles with high risk of bias? 11 (?) of the articles including in this study was scored with a moderate methodological strength, how does that effect your results?
Agreement with other studies and transferability of results.
- Minor- Delete please line -106. Third reviewer (line –117) third author? Correct initials?
Author Response
Reviewer 2: PRISMA-checklist should be included as an additional file.
Response: Thank you, we have included this as an additional file, entitled Appendix B.
Reviewer2: Figure 1. Revise figure 1 according to the PRISMA statement. What about the grey literature?
Figure 1 was developed in accordance with the PRISMA Statement, reflecting the article flow throughout each stage. Our grey literature searches did not yield any included findings and this is reflected in the upper right box of the Figure. To increase clarity, we have adjusted the wording of the upper right box to align with the original PRISMA template and have indicated in brackets that this was grey literature searching. To increase consistency and clarity, we have also increased the font size of the overall PRISMA diagram.
Reviewer 2: Table 1. Huge table. Structure the table. For example, the country name is mentioned twice in the table. The Main-finding should be narrow down to only regarding HRB. Perhaps divide table 1 to two tables to clarify the main findings.
Thank you for these suggestions. We have removed the “study setting” column as the country name is mentioned in the title, as noted. As suggested, we have split Table 1 into Table 1 and Table 2. Table 1 presents the study characteristics, and Table 2 presents the study findings. The original Table 2 (the quality assessment table) is now Table 3. In terms of the main findings column, we think that it is important to retain the findings regarding neighbourhood factors as they provide contextual information to the narrative summary presented in the results section.
Reviewer 2: Appendix A: Search strategy for each database is needed, please also make the appendix clearer, it’s not readable.
We have increased the line spacing of the appendix to make the search strategies more readable. We have added sample searches for each database.
Reviewer 2: Rationale & Objectives: Why is it important to compare HRB urban rural in sub-Sahara? The aim needs to be clearer with one or two research questions. Is it similarities or/and differences?
Thank you for these questions, they will ultimately strengthen the paper. We have added the study’s aim and main research question to be addressed, in addition to the study’s objectives, to the introduction. We have also added in more contextual information to the introduction, addressing why it is important to compare HRBs among adolescents in rural and urban SSA.
Reviewer 2:Methods: Clarify the score of the quality assessment, what does the score 7 means? Risk of bias?
Yes, a score of 7 or above indicates a methodologically strong study with a low risk of bias. We have clarified this distinction in the manuscript.
Reviewer 2:Please describe more about the differences between included studies and excluded studies. Line 177-1978, “fifteen studies reported statistically significant…” of what?
We have added more information about the studies that were excluded and have highlighted the most frequent reasons for exclusion.
We have added a statement clarifying that fifteen studies reported statistically significant (p <0 .05) results on urban and rural dwelling adolescent engagement in HRBs and/or neighborhood level factors associated with adolescent engagement in HRBs.
Reviewer 2:To clarify the study characteristics – a table or diagram would improve the results section.
Thank you for the suggestion. We have split the original Table 1 into two tables: Table 1 now presents only the study characteristics and Table 2 presents the study findings. The original Table 2 (the quality assessment table) is now labelled Table 3.
Reviewer 2:Structure the results section to make it more readable- the synthesis of results.
To increase readability, we have added underlined headers describing the category of health risk behaviour above each paragraph in the results section.
Reviewer 2:Discussion: Summary of results: Line 254-257, this is not seen in the results section or is a reference missing.
We have added the percentage in the results section (58%) and have clarified the wording in the discussion section. Neighbourhood effects influencing sexual behaviour are described in the results section, as well as in the key findings table.
Reviewer 2:Discussion: Limitations. Self-reported-data and HRB and adolescents needs to be discussed more and self-reported data in the included study.
Thank you for this comment. In the limitations section, we have further elaborated the use of self-reporting in the collection of data on adolescent HRBs. We have also added text to the manuscript referencing evidence on the validity and reliability of HRB self-report data among adolescents with considerations to the results of our study.
Reviewer 2:Discussion: Risk of bias. Why did you choice not to exclude articles with high risk of bias? 11 (?) of the articles including in this study was scored with a moderate methodological strength, how does that effect your results?
Great question. There are a limited amount of low quality studies included in the review. Only two of the included studies in the review received quality assessment scores of below adequate. Their scores were one point below the adequate threshold. As such, we chose to include these studies in the review because though they scored below the threshold, we did not assess them to be major threats to the validity of our review and overall the volume of literature was relatively low specific to our research question. In the limitations section of the review’s discussion, we have further acknowledged and elaborated on the effects that the inclusion of low and moderate quality studies may have on the review. In the same section of the paper we have also elaborated on the risk of bias related to self-report data used in included studies.
Reviewer 2:Discussion: Agreement with other studies and transferability of results.
We have made a number of additions to the discussion section of our review, including new references to wider literature and narration on the alignment of the results of our study with wider literature. We have also further elaborated on the transferability of the results of our study to public health programming and policy to address adolescent health and engagement in HRBs.
Reviewer 2: Minor- Delete please line -106. Third reviewer (line –117) third author? Correct initials?
Line 106: It was unclear which sentence needed to be deleted, our best interpretation is that it was referring to the following sentence: “Please see Appendix A for the sample search strategy for Embase, this search string was adapted for the other databases.” We have deleted this sentence.
Third reviewer initials: Thank you for catching this. This refers to the third author. The initials included her middle name, and for consistency we have adjusted the author list name, as well as the author contributions, to also include this middle name initial.
Reviewer 3 Report
This paper is very well written and a pleasure to read. The authors acknowledge the many limitations of the methodology and of the material available and do not inflate the relevance of their findings, which are limited and confirm largely what is known from the literature. It is good, though, to have a review dedicated exclusively to the Sub-sahara African countries.
Author Response
The urban development is presented for South Africa, however it would be good to have a few more comments on the rate of urban development, e.g. is this the same across the country/across socio- economic strata? This would be helpful for the readership to understand the context of their work and how representative is of the entire population.
Response: We have added more information regarding urbanization across the SSA region to the introduction, including regional differences in the rate of urbanization and the proportion of the urban population. For context, we have also added in more information regarding the socio-economic considerations and differences for urbanization across SSA, including increases in informal settlements, such as slums and shantytowns. We have also added socio-economic differences that are tied to urbanization such as income inequality.
The results are presented well against a small number of other countries, however it would be interesting to know if there are similar findings in other countries from southern Africa, or even sub- Saharan Africa? I.e. is this a trend that is specific to South Africa or can be more generally observed across a number of African countries?
Response: The setting of our review is all of sub-Saharan Africa (SSA). In the discussion, we have added more text comparing our findings to existing literature at a regional level in sub-Saharan Africa.
It would be good to have a clearer prioritisation of the future research steps to address the many gaps highlighted by the manuscript.
Response: Thank you for this feedback, it will strengthen the future research section of the paper. We have added three clear priorities for future research which relate to the gaps highlighted by the manuscript.
Round 2
Reviewer 2 Report
- Figure 1. Revise figure 1 according to the PRISMA statement. What about the grey literature? What about records identified through other sources?
- Results section: the new text (line 188-195), sounds like exclusion criteria, that should be in the method sections.
- Reference 43, what is the differences between the two regions?
- Appendix A still needs to be clarified.
Author Response
Reviewer 2’s comments:
- Figure 1. Revise figure 1 according to the PRISMA statement. What about the grey literature? What about records identified through other sources?
Response: Figure 1: Figure 1 was developed in accordance with the PRISMA Statement (reference 39). We did not find any grey literature relevant to our selection criteria which is depicted in the upper right box of the PRISMA diagram (Figure 1). We did identify additional records by scanning reference lists of the included studies, and have presented this in the lower left box to ensure representation of the stage of the search where this was performed. If there are additional aspects this reviewer is looking for please specify.
- Results section: the new text (line 188-195), sounds like exclusion criteria, that should be in the method sections.
Response: Our exclusion criteria are specified in the methods section under section 2.3 “Selection criteria.” The exclusion criteria that are referenced on lines 188-195 are described there. The new text (added after revision round 1) described which of these exclusion criteria were the most frequent reasons for exclusion. To increase clarity, we have added a sentence indicating that these exclusion criteria were a-priori. This is now marked in red in the results section.
- Reference 43, what is the difference between the two regions?
Response: The study setting of reference 43 was in two regions of Northeastern Tanzania: regions of Kilimanjaro and Arusha, both of which contain rural and urban regions. This was our primary focus as our study was concerned with examining and comparing urban and rural adolescent engagement in HRBs, in countries situated in Sub-Saharan Africa. The rural-urban comparisons of adolescent engagement in HRBs within both Kilimanjaro and Arusha are captured in our study results. Similarities and differences between the Arusha and Kilimanjaro regions include: both are mountainous regions with rural and urban areas; the methods section of reference 43 describe Kilimanjaro to have about a population of 1.2 million people and Arusha to have a population of 1.4 million people; ethnic groups varied between regions, where the major Indigenous ethnic groups were described to be Chagga and Pare ethnic in Kilimanjaro and were Maasai, Meru and Iraqw and Arusha.
- Appendix A still needs to be clarified.
Response: Sample search strategies for each database, along with date of search, has been added to appendix A. We have increased the line spacing of the appendix. If there are additional aspects required in Appendix A please specify.